# Oncobiomics: Leveraging Microbiome Translational Research in Immuno-Oncology for Clinical-Practice Changes

**DOI:** 10.3390/biom15040504

**Published:** 2025-03-31

**Authors:** Carolina Alves Costa Silva, Andrew A. Almonte, Laurence Zitvogel

**Affiliations:** 1Gustave Roussy Cancer Campus (GRCC), Clinicobiome, 94805 Villejuif, France; alvescscarolina@gmail.com (C.A.C.S.); andrew.almonte@gustaveroussy.fr (A.A.A.); 2Institut National de la Santé Et de la Recherche Médicale (INSERM) U1015, Equipe Labellisée—Ligue Nationale Contre le Cancer, 94800 Villejuif, France; 3Faculté de Médecine, Université Paris-Saclay, 94270 Kremlin-Bicêtre, France; 4Center of Clinical Investigations BIOTHERIS, INSERM CIC1428, 94805 Villejuif, France

**Keywords:** cancer, gut, immunotherapy, immuno-oncology, microbiota, microbiome, biomarkers, fecal microbiota transplantation, diet

## Abstract

Growing evidence suggests that cancer should not be viewed solely as a genetic disease but also as the result of functional defects in the metaorganism, including disturbances in the gut microbiota (i.e., gut dysbiosis). The human microbiota plays a critical role in regulating epithelial barrier function in the gut, airways, and skin, along with host metabolism and systemic immune responses against microbes and cancer. Collaborative international networks, such as ONCOBIOME, are essential in advancing research equity and building microbiome resources to identify and validate microbiota-related biomarkers and therapies. In this review, we explore the intricate relationship between the microbiome, metabolism, and cancer immunity, and we propose microbiota-based strategies to improve outcomes for individuals at risk of developing cancer or living with the disease.

## 1. Introduction

The human gut microbiota are an ecological niche that largely consists of members from the domain Bacteria, but they are also populated to a lesser extent by members from the domains Archaea and Eukarya (e.g., fungi) [1]. Research over the past decade has also shown they have a profound impact on anti-cancer immune responses. This concept led to the development of “Oncobiomics”, which studies the impact of the microbiota on cancer-immune surveillance. It was developed based on four foundational concepts [2,3]. First, retrospective and prospective studies have demonstrated the negative effect of antibiotics (ATB) on the clinical results of immuno-oncology (I-O) therapies such as immune-checkpoint inhibitors (ICIs) [4,5] and chimeric antigen receptor T cell (CAR-T cell) therapies [6,7]. Second, the shotgun metagenomics sequencing (MGS) of stool samples has enabled the association between gut microbiota composition and microbial metabolic pathways with clinical benefit to ICI across tumor types and geographical regions [3,8,9,10,11,12]. Confounding variables such as geographical specificities, technical factors (e.g., extraction methods), and different definitions of clinical outcomes can contribute to variations for which taxa correlate with therapeutic response between studies [4,13]. However, despite this variability, the prevalence or relative abundance of key taxa such as *Akkermansia muciniphila* and members of the *Lachnospiraceae* or *Oscillospiraceae* families have been repeatedly correlated with improved therapeutic responses [4,14]. Third, fecal microbiota transplantation (FMT) from human donors into mice (patient-derived avatar mice) has established a causal relationship between gut microbiota composition and ICI treatment outcomes. Indeed, oral gavage with specific immunogenic microbes or FMT from treatment-responsive patients could correct resistance to ICI in tumor-challenged mice [10,15,16]. Finally, pioneer studies using microbiota-centered interventions (MCIs) have shown promising results in improving I-O treatment efficacy for patients diagnosed with advanced tumors [17,18,19,20,21,22].

Manipulating the gut microbiota to boost anticancer immune responses is a novel and promising therapeutic strategy, especially against immunogenic cancers like melanoma. In this review, we take readers on a journey through the intricate relationship between the microbiome, metabolism, and cancer immunity, discuss the development of diagnostic tests for gut dysbiosis, and examine microbiota-based strategies to improve therapeutic outcomes.

## 2. Impact of Microbes on Immunosurveillance

Over the past decade, I-O treatments have become essential treatment options for patients with solid and hematological malignancies [23]. The tumor microenvironment (TME) plays a critical role in dictating the efficacy of these I-O therapies. For example, they will be less effective if the TME either physically excludes cytotoxic T cells (i.e., “immune-deserts” or immune-excluded tumors), or has an immune infiltrate dominated by T cells rendered dysfunctional due to interactions with immunosuppressive cells [24]. A better understanding of the factors regulating the TME has led to the development of biomarkers and adjunct therapies aimed at enhancing the efficacy of current I-O treatments [25].

**Impact of antibiotics (ATB)**. ATB leads to reduced microbial (both fungal and bacterial) diversity and a compositional shift dominated by harmful *Enterocloster* spp. and *Hungatella hathewayi* [15,26,27], which negatively impact host metabolism and immunity [26,27]. For instance, our laboratory has demonstrated that the “blooming” of *Enterocloster* spp. following ATB cessation disrupts the expression of ileal mucosal addressing cell adhesion molecule-1 (MAdCAM-1) through perturbation of bile-acid metabolism [26,27]. The downregulation of MAdCAM-1 on the luminal side of high-endothelial venules in the gut mucosa consequently allows the migration of (typically) gut-resident immunosuppressive α4β7+ regulatory Th17 (Tr17) cells towards extraintestinal tumors, consequently promoting an immunosuppressive TME and leading to cancer progression and treatment resistance [26].

Indeed, multiple courses of ATB further downregulate MAdCAM-1 in patients with cancer undergoing ICI treatment [26,27]. Gut microbiota composition and MAdCAM-1 levels do not recover until 90 days after ATB cessation [27,28], emphasizing the need for therapies to accelerate recovery of gut fitness to promote optimal clinical benefit. We demonstrated using preclinical models that *Akkermansia* spp. can prevent the downregulation of MAdCAM-1 [26], thus reestablishing responses to ICI [29] despite gut dysbiosis and showing that microbiota-modulating strategies can restore the MAdCAM-1/α4β7 axis. Furthermore, ATB alters bile acid metabolism, which leads to alterations in MAdCAM-1 expression [26,27]; however, not all bile acids have the same effects [26,30]. Preclinical data show that the conjugated secondary bile acid lithocholic acid (LCA) can decrease the ileal MAdCAM-1 expression, reprogramming the tumor microenvironment [30].

**Cross-reactivity between cancer and bacteria.** The gut microbiota additionally acts as a reservoir of immunogenic molecules that either promote immune cell activation or closely resemble cancer-associated antigens. Lipopolysaccharide (LPS) and other proteins like Amuc_1100 (a protein of unknown function expressed on the surface of *A. muciniphila*) can bind to toll-like receptors (TLRs) and thus influence immune activation [31,32]. Additionally, the carbohydrate polymer microbial polysaccharide A, most notably produced by *Bacteroides fragilis*, stimulates Th1-mediated immune responses [33]. Cross-reactivity between microbial and cancer-derived peptides has also been observed in multiple studies. For instance, Fluckiger et al. used murine models to demonstrate that microbial peptides encoded by the tail length tape measure protein (TMP) of a prophage found in *Enterococcus hirae* can elicit CD8^+^ T cell-mediated immunity against cancer through MHC class I molecules [34,35]. Another study demonstrated this concept in a clinical context by showing that certain tumor-infiltrating lymphocytes (TILs) and TIL-derived T cell clones respond to both glioblastoma neoantigens and gut microbiota-derived peptides, further indicating molecular mimicry and cross-reactive immune recognition between gut bacteria and cancer [36]. Collectively, these findings support the argument that a highly diverse gut microbiome may enhance immune surveillance by increasing the potential for cross-reactivity between microbial peptides and tumor-associated antigens, including neoantigens. However, cross-reactivity between gut microbial antigens and tumor neoantigens suggests a similar potential between gut microbes and autoantigens [37,38]. Such interactions could increase the incidence of immune-related adverse events (IrAEs) following ICI therapy and require further study.

**Microbial metabolites.** Microbes can further impact immunosurveillance and anticancer immunity by influencing a patient’s metabolic health. Microbial-related metabolites such as polyamines and inosine influence T cell immunity [39,40], restore tissue-resident memory T cells [41], and enhance fatty acid oxidation that leads to CD8^+^ T cell activation [42]. The gut microbiota produces several metabolic products, such as vitamins (e.g., riboflavin, niacin, pantothenic acid, pyridoxine, and cyanocobalamin) that favor normal tissue function and immunosurveillance [43,44,45]. Short-chain fatty acids (SCFAs) are another important metabolic byproduct. They are produced through the anaerobic fermentation of non-digestible fibers [1] and act as an energy source for intestinal epithelial cells and signaling molecules in the periphery [46]. They are also known to improve memory CD8^+^ T cell function [47,48].

The relationship between microbial metabolites and immune function underscores the importance of anatomical sites in shaping immune responses. Various studies have highlighted the key role of the ileum in regulating immunosurveillance [49,50,51,52,53], not only due to the ileal microbiota composition (which differs from colonic fecal samples [1]) [26,53] but also antigenic peptides released by enterocytes [49]. For example, Roberti et al. demonstrated that higher rates of apoptotic ileal enterocytes were associated with improved outcomes in patients with colorectal cancer (CRC) located on the right side (ascending colon) treated with chemotherapy regimens based on oxaliplatin [49]. We hypothesize that the ileal microbial biology may be implicated in the ICI benefit [54,55] of patients with germline susceptibility to CRC (mostly right-sided tumors) [49,51,56].

**Intratumoral microbiota.** Several studies have confirmed the presence of microbiota within tumors, though at very low levels [57,58,59,60,61,62,63,64,65]. The origin of these bacteria is still under investigation and may be context-dependent. However, we know that microbes can translocate from the gut lumen by invading the mucosal barrier or via hematogenous dissemination to lymph nodes or tumors, influencing tumor biology [66,67] and potentially the efficacy of I-O treatments [47]. Murine models have demonstrated that ICI treatment can induce microbial translocation via dendritic cells (DCs) into mesenteric lymph nodes (mLN) and tumor-draining lymph nodes (TdLNs), promoting anticancer responses [68]. Furthermore, ATB decreased DC-mediated microbial translocation from the gut into tumors, resulting in reduced DC and effector CD8^+^ T cell responses and promoting ICI resistance [68]. Fluckiger et al. also found that TMP+ *E. hirae* can translocate to the mLN and spleen and improve the outcomes of both cyclophosphamide and anti-PD-1 treatments [34]. Intratumoral microbiota can be characterized through various approaches, including sequencing, pathology, and culture methods [69].

Although sequencing methods are the gold standard in clinical research, the method of choice ultimately depends on the scientific question. Moreover, different methods can be complementary [69]. For instance, sequencing can identify and quantify taxa within a sample (e.g., how much of taxon A is present?). This approach enables the alignment of microbial sequences with publicly available genomes and supports the use of isolated species in preclinical and functional experiments. On the other hand, culturomics enables the isolation of live microbial colonies from tumors and surrounding tissues (e.g., is taxon A present in the sample and where?). This method allows us to determine where in a given sample the bacteria are located—tumor or adjacent healthy tissue—and retain the clinical isolates for further study.

Recent studies have increased our understanding of the Janus face of the human microbiome, revealing its role in complex and multifactorial processes. The intratumoral microbiota may increase the metastatic potential of cancer cells through genomic instability, a hallmark of cancer [66,70]. The presence of intratumoral *Fusobacterium nucleatum* is correlated with an immunosuppressive TME in CRC [71,72], lung cancer [73], and breast cancer [74], leading to decreased infiltration of CD8^+^ T cell and reduced cytotoxic activity [71,72,74]. Furthermore, colibactin is a carcinogen produced by pks+ *Escherichia coli* strains and their presence in CRC has been associated with disease onset and progression [75,76]. Thus, different bacterial species and strains can have distinct—and sometimes detrimental—effects on disease progression and treatment outcomes.

Although therapies targeting cancer-associated microbial neoantigens are promising therapeutic or prophylactic strategies [77,78], there is little evidence suggesting that intratumoral bacteria are clinically relevant. Regardless, two studies have demonstrated that *Escherichia* spp. can be found in lung and bladder cancer biopsies and surgical specimens [79,80]. Moreover, intratumoral *Escherichia* spp. was associated with superior survival in patients with non-small cell lung cancer (NSCLC) treated with ICI monotherapy (but not in combination with chemotherapy regimens) [80]. Interestingly, patients with urothelial carcinoma treated with neoadjuvant PD-1 blockade who mounted humoral responses against *E. coli* had better prognoses [79]. It is known that specific humoral responses against microbes are crucial to coordinating the host microbial homeostasis and preventing or controlling pathogen outgrowth and systemic infections [81]. Liu et al. showed that targeting the tolerogenic intratumoral *F. nucleatum* either via a bacterial outer membrane vesicle-coated nanoplatform or metronidazole, a nitroimidazole-class ATB, may increase the TME immunogenicity in patients with triple-negative breast cancer (TNBC) [74]. Similarly, liposomes loaded with ATB targeting harmful anaerobic bacteria drove microbial-specific T cell immunity in CRC models [78]. These studies highlight the clinical prospects of modulating the intratumoral bacterial composition or inducing specific immune responses towards clinically relevant microbes to improve cancer treatment outcomes [82,83].

Leveraging the microbiota can enable the development of robust interventions to overcome the negative, or enhance the positive, microbial-related factors implicated in the cancer-immune set point [84]. We and others aim to improve the clinical results of I-O treatments and the quality of oncology clinical care by reestablishing gut fitness and promoting more effective anticancer immune responses [4].

## 3. Development of Diagnostic Tools

Microbiome sequencing has enabled researchers to demonstrate the association between gut dysbiosis and human diseases, including cancer [85,86]. Indeed, patients with cancer exhibit hallmarks of cancer-associated stress ileopathy, characterized by anatomical and functional changes in the ileum caused by β-adrenergic receptor signaling-mediated stress, consequently leading to increased gut permeability and protracted gut dysbiosis dominated by Gram-positive microbes (e.g., *Enterocloster* spp. described above) [53]. Gut dysbiosis is associated with worse clinical outcomes and primary treatment resistance across various cancers and treatment types, notably I-O regimens [14,87]; patients are more likely to benefit from I-O treatments if they harbor health-related gut commensals at the baseline [14,37,88]. Additionally, gut microbiota composition has been correlated with toxicity to both CAR-T cell therapies [6] and ICI [89,90], specifically the incidence and severity of IrAEs. Moreover, dysbiosis has been characterized by metabolic disruptions, notably reduced production of SCFAs, elevated levels of inflammatory cytokines, and compromised gut barrier integrity. Importantly, we have shown that microbiota and metabolic profiles may act as confounding factors that need to be considered in randomized clinical trials to minimize selection bias and its potential to impact the clinical outcomes of investigational therapies [91,92].

We, together with our colleagues, have been working to discover taxonomic biomarkers to identify patients who are dysbiotic (Figure 1). Recent studies demonstrated that various cancers share consistent MGS signatures with diverse inflammatory or metabolic diseases, such as cardiovascular disorders, type 2 diabetes, inflammatory bowel disease, irritable bowel syndrome, and mental health conditions including depression [13]. In a Dutch study comprising 8,208 individuals across three generations, Gacesa et al. compared over 20 cases for each of 81 different diseases, including those just mentioned, and found they were associated with decreases in *Faecalibacterium*, *Bifidobacterium*, *Butyrivibrio*, *Subdoligranulum*, *Oxalobacter*, *Eubacterium*, and *Roseburia* genus. In contrast, MGS patterns comprised of *Anaerotruncus*, *Ruminococcus*, *Bacteroides*, *Holdemania*, *Flavonifractor*, *Eggerthella*, and *Enterocloster* genus were positively associated with those same diseases [13]. These findings are notable because the shared microbiome signatures were found across several unrelated maladies. Similarly, a pan-cancer meta-analysis found evidence that patients who benefit from ICI were enriched with members of the *Lachnospiraceae* and *Oscillospiraceae* families members (notably containing the *Faecalibacterium* genus) and *Akkermansia* spp. [14]. However, patients experiencing resistance to ICI experienced an overrepresentation of the *Enterocloster* genus in addition to pro-inflammatory oral taxa (e.g., *Streptococcus* spp.), and members of the *Veillonella* and *Hungatella* genus [14]. Therefore, developing diagnostic tools to identify dysbiosis in cancer patients can be translated to other diseases (Figure 1).

## 4. Fecal Markers

The presence of fecal *A. muciniphila* before starting an ICI treatment regimen correlates with improved clinical results in patients with NSCLC and renal cell carcinoma (RCC) [10,29,83,93]. However, correlations with treatment outcomes cannot always be as straightforward as relying solely on relative abundance or prevalence [10,29,93]. For example, a prospective study demonstrated that trichotomizing the relative abundance of the *A. muciniphila* species-level genome bin (SGB) 9226—categorizing it as abnormal values if either absent or its relative abundance is greater than 4.799%—provided a better stratification of patients who are likely to respond to treatment (*A. muciniphila* SGB9226 is a surrogate marker of gut barrier fitness) [29,83]. The absence of *A. muciniphila* or its excessively high abundance correlates with poorer overall survival (OS), an immunosuppressive gut microbiota composition and higher host exfoliate transcriptome (a surrogate marker of stress ileopathy). In contrast, *A. muciniphila* values within a healthy range correlate with a health-related gut microbiota composition, a lower host exfoliate transcriptome, and Th1 polarization in the TME [29,83]. Interestingly, immune responses directed against *Akkermansia* spp. leading to bacteria eradication may be deleterious for patients with NSCLC [82,83]. *Faecalibacterium prausnitzii* is another highly immunogenic bacterium in the gut microbiota of healthy individuals and a known butyrate producer [94,95], and its decreased abundance is associated with poorer clinical outcomes across cancer types [8,91,94].

Notably, a single microbial species can have multiple SGBs due to factors such as genetic diversity and strain variation [96]. Thus, not all SGBs may have the same impact on human health and clinical benefit to I-O strategies [91].

Due to the limitations in predicting clinical outcomes across cohorts and cancer types using isolated MGS species, we and others have focused on the development of genome- and community-based approaches that derive individual metrics that can be used for diagnosing gut dysbiosis. These efforts allow opportunities for selecting appropriate recipients of relevant MCIs and any donors that may contribute toward these therapies [88,97,98,99,100]. The Gut Microbiome Health Index (GMHI), a predictive index based on a mathematical model using 50 MGS species, allowed Gacesa et al. to significantly differentiate individuals with self-reported health versus disease in their cross-generational Dutch cohort [13,97]. Moreover, two competing microbial ecological functional units that collectively increase or decrease in abundance (e.g., “guilds”) are composed of different taxonomic backgrounds and predict the metabolic syndrome of patients with type 2 diabetes [98,99].

Likewise, we performed an ecological distribution of fecal MGS species into two opposing species interacting groups (“SIGs”) with disparate predictions for OS of patients with cancer after 12 months of treatment [88]. Using the whole-population-based network as a foundation, we developed an individual-level numerical score called the topological score (TOPOSCORE) [88], which takes into account the relative abundances of 37 harmful (SIG1) and 45 beneficial (SIG2) MGS, along with the trichotomized relative abundance of *A. muciniphila* SGB9226 described earlier. Notably, SIG1 bacteria are associated with MGS metabolic pathways involving the β-oxidation of fatty acids, the degradation of 5′deoxyadenosine, L-phenylalanine, purine, and L-histidine, and the biosynthesis of guanosine and L-lysine. In contrast, SIG2 bacteria correlated with polyamine and tryptophan pathways, which are associated with T-cell fitness. The translation of the MGS-based score into a qPCR-based score—distilled to a clinically relevant list of 21 bacteria—creates opportunities for rapid and simple clinical implementation [88].

The leverage of large datasets identifying health- and disease-related MGS signatures [13] opens new avenues for identifying robust fecal markers as diagnostic tests of dysbiosis that can be applied in different settings. In addition to selecting patients who will benefit from MCI upon starting ICI therapy, MGS signatures hold the potential for determining the optimal ICI treatment duration. A prospective study is evaluating whether patients with persistent dysbiotic microbiota might benefit from discontinuing ICI regimens after two years without disease progression [101].

MGS signatures can additionally be used to identify individuals at risk of developing diseases like cancer. For instance, patients with CRC have distinct MGS signatures from those with adenomas or healthy individuals [102,103]. The MGS signature associated with CRC was marked by an increased relative abundance of specific subclades of *F. nucleatum*, which may have varying biological significance [72]. For example, patients with CRC had decreased levels of *Ruminococcus bicirculans* and *F. prausnitzii* subclades [102,103]. Furthermore, *H. hathewayi* and *Methanobrevibacter smithii* were associated with stages III–IV or IV CRC, respectively [103].

Similarly, five distinct fecal host-derived micro RNAs (miRNAs) distinguished patients with CRC from individuals without CRC in a multicentric study [104,105]. Interestingly, Mediterranean dietary intervention modulated pro-inflammatory fecal miRNAs [104,106,107,108], showing that a longitudinal transcriptome evaluation of miRNAs may help assess the responses to MCI.

The current challenge relies on the prospective validation of these fecal markers across countries and cancer types towards implementation in clinical routine.

## 5. Circulating Markers

Advances in our understanding of underlying mechanisms governing the gut-immune axis have enabled the discovery of blood-based surrogate markers for gut dysbiosis. For example, we have demonstrated that soluble MAdCAM-1 (sMAdCAM-1), measured via a Luminex assay, correlated with ileal transcripts and the clinical prognosis of cancer patients [26,109]. Moreover, sMAdCAM-1 levels correlated with gut microbiota composition, showing that it can be used as a biomarker of gut dysbiosis [26]. Patients with low levels of sMAdCAM-1 have a poorer prognosis and a gut microbiota composition enriched with tolerogenic *Enterocloster* spp., *Veillonella* spp., and *Thomasclavelia ramosa* (formerly known as *Erysipelatoclostridium ramosum*) [26].

Lymphocytic responses against gut microbes can also be potential biomarkers and can be easily assessed via flow cytometry or microbial-specific T cell stimulation assays [79,82,83,110]. Goubet et al. demonstrated that the titer of IgG specific to uropathogenic *E. coli* can be a potential biomarker associated with the clinical benefit of ICI treatment regimens in patients with bladder cancer [79,110]. Specifically, patients who had higher titers of anti-*E. coli* IgG had a better prognosis following treatment [79,110]. In another study, NSCLC patients with high titers of anti-*Akkermansia* IgG were more likely to lack *A. muciniphila* in their fecal microbiota, a marker of a tolerogenic gut microbiota composition, and experienced worse clinical outcomes [83]. Furthermore, the development of memory T cell responses against distinct microbes may be clinically relevant to identifying patients who could potentially benefit from ICI, such as *Akkermansia*-specific follicular helper T cell responses [82]. Immune responses against gut commensals deserve further investigation in prospective MCI clinical trials, both as baseline biomarkers for treatment success and to assess prospective responses to MCIs, such as the evaluation of enterofecal compatibility before receiving an FMT. Indeed, Thomas et al. were able to use murine models to demonstrate distinct impacts on immune responses, especially regarding RORγt+ T cell and macrophage populations, and host metabolism following different patterns of FMT engraftment [111]

Distinct circulating metabolites appear to be promising hallmarks of cancer prognosis and the response to ICI [91,112,113]. However, there is no consensus in the literature on which metabolomic signatures are most effective in I-O, primarily due to technical variation across studies, such as sample collection time, sample type, and storage methods. In a cohort of patients with locally advanced melanoma, an early metabolic shift was detected before the clinical diagnosis of cancer recurrence. This shift was characterized by an accumulation of acylcarnitine chains, fatty acids, and carboxylic acids, which were associated with relapse [91]. Importantly, anticancer and concomitant therapies may impact the interplay between host, microbiome, and metabolism [27,91] and, therefore, the clinical prognosis, thus highlighting the importance of the longitudinal assessment of these parameters to guide clinical decisions. Desaminotyrosine (DTA), a flavonoid metabolic byproduct, improves gut barrier fitness and modifies the TME into a more immunogenic phenotype. It is, therefore, associated with improved ICI treatment outcomes and also overcomes the deleterious effects of ATB [112,114,115]. Alongside other fecal metabolites, such as SCFAs, isovaleric acid, and indole-3-carboxaldehyde, fecal DTA contributes to the immunomodulatory metabolites risk index (IMM-RI) [116], an approach analogous to fecal MGS scores. Patients who are considered IMM-RI^low^ were associated with improved clinical outcomes (as measured by survival, rate of relapse, and transplant-related mortality) after allogeneic hematopoietic stem cell transplantation [116].

Ideally, blood tests would be used as proxies for gut dysbiosis and represent promising alternatives to time- and cost-demanding fecal sequencing tests, but they must be clinically validated before widespread adoption in oncological practice.

## 6. Promising Microbiota-Centered Interventions (MCIs) in the Immuno-Oncology (I-O) Field

MCI strategies aim to regulate the gut microbiota and restore a healthier, more immunogenic microbial composition. Examples of MCI include FMT, prebiotics, probiotics, live biotherapeutic products (LBPs), diet, and concomitant medications [4]. Current evidence from clinical trials shows that MCI may restore host–microbe homeostasis through changes in the gut microbiota composition, metabolism, and immune system, thereby favorably altering the TME and leading to improved ICI treatment outcomes.

### 6.1. Comedications

Given the potential influence of comedications on the microbiota–cancer–immunity axis (Figure 2), microbiome research supports recommendations to discourage polypharmacy and promote their rational use. ATB [4,5,117,118,119] and proton pump inhibitors (PPIs) [120,121] are associated with reduced efficacy of ICI treatments. This highlights the importance of promoting good prescribing practices and regularly re-evaluating maintenance therapies to discontinue unnecessary comedications in patients without a clear indication. When clinically recommended, we should also consider the timing of ICI administration and the spectrum of action of ATB and PPIs [27,121].

A large meta-analysis of ICI-treated NSCLC patients has demonstrated that the deleterious effect of ATB intake occurs between 60 days before and 42 days after ICI treatment initiation [4]. This finding was validated by a second, larger meta-analysis [5] and a prospective cohort study investigating the immunosuppressive effects of ATB in patients with NSCLC, RCC, or bladder cancer undergoing ICI-based treatments [27]. Notably, ATB seem to have differing effects across treatment classes; while they also impact the efficacy of tyrosine kinase inhibitors (TKIs) [122,123], no such effect was observed with chemotherapy regimens [124].

Narrow-spectrum ATB does not impact survival [27,121]; indeed, the targeted elimination of harmful microbes may even be beneficial [53,78]. We demonstrated that vancomycin eliminates *Enterocloster* spp. and reduced markers of cancer-related ileopathy and tumor growth in preclinical models [53]. Moreover, patients who received ATB targeting *F. nucleatum* (a bacterium associated with disease progression in CRC), such as nitroimidazoles or lincomycin-class compounds, showed improved disease-free survival compared to those who received no ATB or ATB with a different spectrum [78], provided the primary tumor was still in situ. Additionally, a liposome-formulated ATB was designed to specifically target intratumoral bacteria-induced T cell immunity by generating cancer-specific microbial neoantigens through bacterial elimination [78]. Consistent with these results, a randomized phase 2 clinical trial (NCT04264676) has been designed to evaluate the use of metronidazole or placebo control concomitant with chemotherapy in CRC patients who will be selected based on qPCR confirmation of *F. nucleatum* in colon tissue biopsies.

ATB and PPIs are not the only comedications that influence the gut microbiota in cancer patients. Preclinical models indicate that cancer-induced β-adrenergic stress precipitates intestinal dysbiosis, mucosal atrophy, and shifts in inflammatory metabolic pathways [53]. Consequently, we propose that β-blockers (e.g., propranolol) may provide a promising strategy for mitigating these dysbiosis-related complications. However, there is conflicting evidence between studies regarding the impact of blocking the dominance of stress ileopathy sympathetic signaling [53]. While β-blockers have been associated with positive outcomes in patients with TNBC [125,126], they appear to have no benefit for patients diagnosed with RCC [127,128]. Thus, further research is required to refine comedication strategies designed to enhance treatment outcomes or ameliorate disease symptoms.

### 6.2. Fecal Microbiota Transplantation (FMT)

Pioneer clinical trials evaluating FMT have demonstrated the feasibility, safety, and efficacy of this method in patients with solid tumors, despite variability between trial designs (Table 1) [17,19,22,129]. Fecal material was collected from either treatment-responsive patients with cancer who achieved persistent complete (CR) or partial (PR) responses to ICI [17,18,19] or from heavily screened healthy volunteers (HVs) [22]. FMTs were performed via endoscopy [17,19], oral capsules [22], or both [18] and given either after disease progression following an initial ICI treatment [17,19,129] or before or concomitant with first-line ICI treatments [18,22]. In seminal studies led by Davar and Baruch, administering FMT to treatment-refractory patients before ICI therapy restored treatment efficacy—as measured via complete and partial responses—in 20–30% of patients with advanced melanoma [17,19]. Park et al. further showed that FMT given before resuming ICI treatments led to a disease control rate in 46% of patients with digestive tumors who had primary or secondary resistance to ICI, with additional data demonstrating immune activation following FMT [129]. Lastly, upfront FMT appears to enhance outcomes in first-line ICI therapy, as evidenced by comparisons with pivotal trials [22] or standard-of-care treatment arms [18].

FMT trials highlight the clinical relevance of the ecological taxonomic distance between the fecal microbiota of FMT donors and recipients, in addition to the engraftment of relevant microbial species [17,19,22]. Patients who benefit from FMT experience a gut microbiome shift toward their donor’s gut microbial composition [17,19,22]. While switching from tolerogenic (such as *Enterocloster bolteae*) towards immunogenic bacteria (including several *Ruminococcaceae*), these patients showed improved markers of gut fitness such as circulating ST2 (a marker of gut permeability [53]) and propionate (beneficial bacteria-derived SCFA) [22]. Interestingly, the development of specific IgG responses against donor-specific microbes was associated with efficient colonization of donor microbiota and improved response to ICIs [19].

FMT is a straightforward means of rapidly and drastically changing the gut microbiota composition of a given individual. However, the need for specialized centers to handle quality control standards, in addition to the poor stability and reproducibility of fecal material, makes the routine use of this method difficult [130,131]. Leveraging the microbiota ecological dynamics that shape the recipient’s microbiome after an FMT procedure is an important step toward more reliable and replicable interventions.

**Table 1 biomolecules-15-00504-t001:** Clinical trials evaluating fecal microbiota transplantation in immuno-oncology for solid tumors.

Study	Tumor	Setting	Protocol	N	Arms	Control Arm	Admin. Route	Donor (N)	Preparation Method	N FMT	Primary Endpoint	mFU	mPFS (mo)	mOS (mo)	ORR (%)	DCR (%)	DoR (mo)
SMC17-3956 [17]	Melanoma	Rescue	ICI	10	1	NA	Colonoscopy and oral capsules	Pts with CR for over a year (2)	ATB for 3 days + BCS	Colonoscopy on D0, oral capsules on D1 and D12 × 6 cycles	FMT/ICI-related AEs	113 days	NR	NR	30	30	All FMT R had PFS > 6 mo
NCT03341143 [19]	Melanoma	Rescue	ICI	15	1	NA	Colonoscopy	PR > 24 mo or CR > 12 mo (7)	NR	1 *	Whether FMT can convert NRs to Rs	7 mo	3	7	20	33	Between 3 and 27 ^£^
NCT03772899 MIMic [22]	Melanoma	Upfront	Anti-PD-1	20	1	NA	Oral capsules	HV (3)	BCS	1	Safety	21 mo	NRe	NR (16 pts were still alive at FU)	65	75	12
NCT04951583 FMT-LUMINate [132,133]	Melanoma	Upfront	Anti-PD-1 + anti- CTLA-4	20	1	NA	Oral capsules	HV (6)	BCS	1	ORR	7 mo	NR	NR	75	75	Between 3 and 12 ^£^
	NSCLC	Upfront	Anti-PD-1	20	1	NA	Oral capsules	HV (6)	BCS	1	ORR	7 mo	NR	NR	75	90	Between 3 and 9 ^£^
NCT04264975 [129,134]	Solid cancers (GI tumors)	Rescue	Anti-PD-1	13	1	NA	Colonoscopy	CR/PR ≥ 6 mo (6)	ATB for 5 days °	1 *	ORR	NR	NR	NR	8	46	NR
NCT04758507 TACITO [18,135]	RCC	Upfront	Anti-PD-1 + TKI	50	2 (R 1:1)	Placebo (saline solution)	Colonoscopy and oral capsules	Long-term R > 12 mo to ICI (1)	NR	3	PFS rate at 1 year	28 mo	14 (vs. 9)	NRe (vs. 25)	52 (vs. 28)	90 (vs. 72)	NR

Administration, admin.; adverse events, AEs; antibiotics, ATB; bowel-cleansing solution, BCS; complete response, CR; Day 0, D0; Day 1, D1; Day 12, D12; disease control rate, DCR; duration of response, DoR; gastrointestinal, GI; healthy volunteers, HVs; immune-checkpoint inhibitors, ICIs; Median follow-up, mFU; median progression-free survival, mPFS; median overall survival, mOS; microbiota-centered intervention, MCI; months, mo; non-responders, NRs; non-small cell lung cancer, NSCLC; not applicable, NA; not reached, NRe; not reported, NR; number, N; overall response rate, ORR; patients, pts; partial response, PR; randomization, Rs; responders, Rs; tyrosine kinase inhibitors, TKIs; versus, vs.. * NCT03341143: one recipient received a new FMT after ATB use due to soft-tissue infection. NCT04264975: subsequent FMT were allowed (same or different donor). ° Before 1st FMT. ^£^ Reported individually.

### 6.3. Next-Generation Probiotics (NGP) and Live Biotherapeutic Products (LBPs)

Research groups are working hard to identify optimal microbial strains or consortia to boost immune responses following cancer therapies [11,29,136]. Early clinical trials have evaluated the LBPs containing a strain of *Clostridium butyricum* MIYAIRI 588 (CBM588) concomitant to first-line ICI combinations (Nivolumab plus Ipilimumab or Cabozantinib) in patients with advanced RCC [20,21] and chemoimmunotherapy for NSCLC [137,138,139]. Although the primary endpoint of the RCC-based studies was not met (i.e., a change in the relative abundance of fecal *Bifidobacterium* spp.), CBM588 was associated with improved objective response rates (ORR) and progression-free survival (PFS) for both cancer types. Notably, CBM588 most significantly improved the PFS and OS of NSCLC patients who had received ATB or PPI before treatment over those who did not [137,139], thus suggesting a role for this LBP in correcting comedication-induced dysbiosis. Preclinical studies have additionally revealed the clinical benefit of *A. massiliensis* strain p2261 (previously classified as *A. muciniphila*) [29] and *F. prausnitzii* strain EXL01 [94] in tumor-bearing mice treated with ICI, paving the way to launch early-phase clinical trials. A phase 2 trial (NCT05865730) assessed the clinical efficacy of Oncobax^®^-AK, a lyophilized and encapsulated strain of *Akkermansia*, in select *Akkermansia*-negative (based on a qPCR test), treatment-naïve RCC and NSCLC patients who are eligible for ICI treatments. The preliminary results presented at the Society for ImmunoTherapy of Cancer (SITC) 2024 by Lisa Derosa showed that Oncobax^®^AK is a safe and improved marker of gut barrier fitness. The phase 2 study conducted by the GERCOR research group (NCT06253611) assessed the efficacy of combining EXL01 with the FOLFOX chemotherapy regimen (folinic acid, fluorouracil, and oxaliplatin) and ICI for patients with advanced gastric cancer expressing a PD-L1 combined positive score (CPS) ≥ 5. ORR is the primary endpoint for both studies. Another early-phase clinical trial is currently underway to evaluate a microbial consortium, Microbial Ecosystem Therapeutic 4 (MET4), combined with ICIs in patients with solid tumors [136]. Although the primary endpoint is not met (microbial ecological outcomes), MET4 is safe and well-tolerated [136].

Importantly, care providers must exercise vigilance in this area, particularly given the widespread commercialization of probiotic products. Spencer et al. demonstrated the deleterious impact of self-medication with commercially available probiotic products [140], highlighting the need for the refinement of robust NGP and LBPs [141]. Furthermore, commercially available *Bifidobacterium*- or *Lactobacillus*-based probiotics negatively impacted the systemic immunity of treated mice, which led to ICI resistance [140]. Thus, more work and clinical validation must be conducted to identify which microbial strains are beneficial and, at an absolute minimum, not deleterious to patient health.

### 6.4. Diet, Metabolites, and Prebiotics

Environmental factors surpass genetics in shaping the gut microbiota [13], thus pointing to dietary strategies as important regulators of gut barrier fitness and host–microbiota interplay [142]. Spencer et al. demonstrated that a high-fiber diet (HFD) is associated with richer commensalism and improved prognosis in ICI-treated patients with melanoma [140]. Every 5 g increase in dietary fiber intake reduced the risk of cancer progression or death by 30% after adjustment for clinical variables [140]. Jiang et al. demonstrated that an HFD intervention influences both the gut microbiome and host metabolism by increasing systemic acetate levels, altering omega-3 and omega-6 polyunsaturated fatty acid profiles, and modulating indole and tryptophan metabolism, particularly in individuals with the lowest baseline fiber intake [143,144]. The randomized phase 2 DIET study (NCT04645680) further demonstrated that HFD is well tolerated, though clinical outcomes results are still pending [143]. Other randomized trials will evaluate the HFD strategy in different settings, such as additional tumor types (patients with NSCLC, NCT05805319) and associated with other lifestyle strategies (e.g., exercise, NCT06298734).

Link et al. provided evidence of personalized nutritional intervention decisions to prevent and treat cancer using the principles of nutrition–immunity interplay [145]. Different dietary strategies (vegan versus ketogenic diets) lead to the upregulation of different pathways and enrichment in immune cells [145]. The ketogenic diet significantly influenced the adaptive immune system and microbial pathways, while a vegan diet mostly impacted the innate immune system. Notably, two weeks of controlled dietary intervention is sufficient to significantly and divergently impact host immunity [143,145]. Ketogenic strategies, either through the ketogenic diet or supplementation with ketone bodies, have shown benefit in preclinical models of tumor-bearing mice treated with ICI regimens [146,147], though results from clinical trials are still pending.

It should be noted that there is the potential for low compliance and significant discontinuation rates in controlled feeding studies [143]. If dietary intervention is to be pursued, cultural differences, socioeconomic status, taste preferences, tumor types, and prescribed anticancer therapies must also be considered to improve patient adhesion and clinical benefit to dietary interventions.

Prebiotics are non-digestible food ingredients that selectively stimulate beneficial gut bacteria, thereby enhancing metabolic functions such as the production of SCFAs and modulation of inflammatory cytokine profiles [148,149]. These changes can improve immune function and may help create a more favorable TME for I-O therapy. Recent research suggests that such microbial modulation is critical for maximizing the efficacy of ICIs. For instance, Messaoudene et al. demonstrated that the prebiotic camu-camu berry (*Myrciaria dubia*), a reddish purple, cherry-like fruit from the Amazon rainforest in South America, exhibits antitumor activity and can overcome resistance to ICI in murine models through effects mediated via the polyphenol castalagin [150]. A phase 1 clinical trial (NCT05303493) is underway to evaluate the effects of camu-camu capsules on standard ICI regimens in patients with advanced NSCLC or melanoma. Ultimately, integrating prebiotics into cancer care could help mitigate IrAEs while enhancing therapeutic responses.

## 7. Discussion

The composition of the gut microbiota impacts cancer immunosurveillance, and leveraging the complex dialogue between microbiota, metabolism, and cancer immunity opens many possibilities for clinical application. However, gut microbiota signatures show limited reproducibility across different datasets, with variations in taxonomic composition between cohorts and countries [151,152]. The lack of standardization in microbiome-sequencing procedures, geographical factors, and different definitions for clinical outcomes may contribute to this limited reproducibility [4,13]. Therefore, we advocate for establishing an international consensus in the microbiome research field to standardize protocols and address this problem [131,153].

Rather than relying solely on the classical Linnaean taxonomy and microbial strains, the alignment and deeper analysis of sequencing data have allowed for significant refinements in microbial classification, as demonstrated by the large-scale metagenomic assembly uncovering over 150,000 microbial genomes in diverse populations [96]. Importantly, distinct strains or subclades of microbial species may have different biological effects and clinical relevance [34,72,91]. Some studies discussed here provide evidence supporting a focus on microbial ecological communities as a whole, rather than on specific microbial subclades [88,98,99]. Whether the use of microbial consortia will bring more clinical benefit than monoclonal NGP or LBP intervention is still an open question.

Other key questions remain to be addressed in the field of MCIs, such as determining the most effective type of intervention (e.g., dietary modification versus FMT) and identifying the optimal patient populations and treatment settings (e.g., rescue versus upfront therapy). The lessons learned in I-O may be transposed to other fields; evidence supports shared MGS signatures between cancer patients and patients with other metabolic and inflammatory diseases [13], suggesting common mechanisms for how the microbiota influences human health [85]. Since a given strategy or approach may not be suitable for every patient, we should consider relying on microbiota-related biomarkers to better evaluate and address individual patient needs. Moving forward, efforts should focus on validating and standardizing diagnostic tests for gut dysbiosis (Figure 1) and conducting more controlled trials with MCI to further explore their synergistic antitumor effects.

## Figures and Tables

**Figure 1 biomolecules-15-00504-f001:**
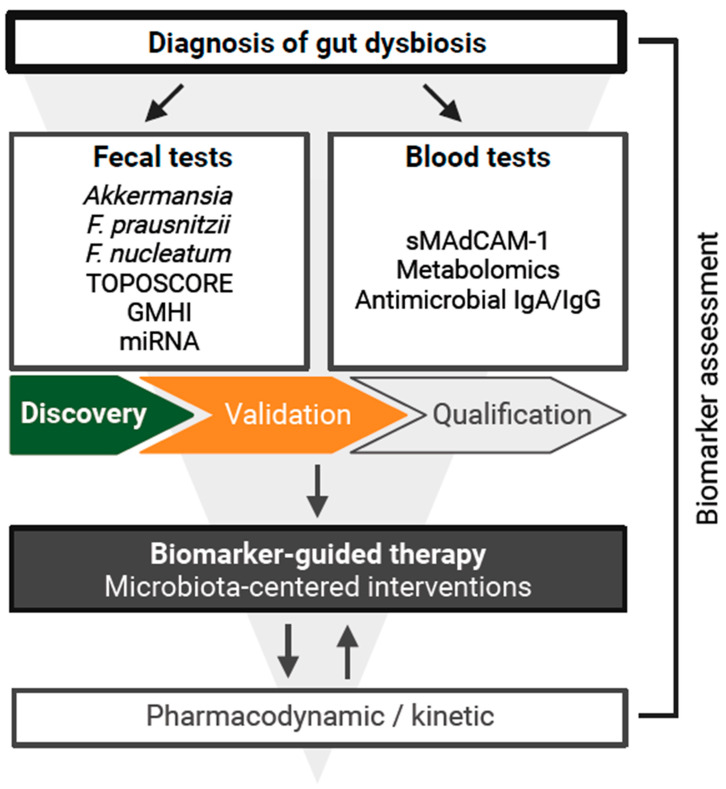
A pipeline for developing biomarker-guided interventions in the microbiota field. We illustrate how clinicians might diagnose gut dysbiosis and inform therapy through fecal or blood-based markers. Fecal biomarkers include key bacterial taxa (e.g., *Akkermansia*, *Faecalibacterium prausnitzii*, and *Fusobacterium nucleatum*), the TOPOSCORE, the Gut Microbiome Health Index (GMHI), and miRNA signatures. Blood biomarkers include soluble mucosal addressing cell adhesion molecule-1 (sMAdCAM-1), metabolomic profiles, and antimicrobial IgA/IgG. Following the discovery, validation, and qualification phases, these biomarkers guide microbiota-centered interventions, with feedback from pharmacodynamic and kinetic assessments to refine treatment strategies.

**Figure 2 biomolecules-15-00504-f002:**
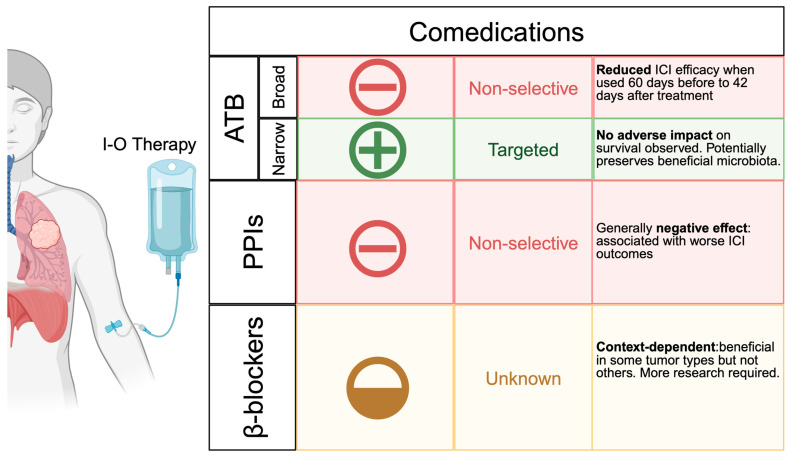
Comedications influencing immune-oncology (I-O) therapy efficacy. Broad-spectrum (non-selective) antibiotics reduce immune-checkpoint inhibitors (ICI) response when administered near treatment, whereas targeted (narrow-spectrum) agents appear relatively safe. Proton pump inhibitors (PPIs) are similarly associated with poorer ICI outcomes, highlighting the need for careful prescribing. β-blockers exhibit context-dependent effects, showing potential benefits in certain malignancies but an uncertain impact in others.

## Data Availability

No new data were generated or analyzed in this study. This review article is based on previously published studies and publicly available literature, all of which are cited in the references section. Data sources referenced in this manuscript can be accessed through the original publications or repositories as indicated in the respective citations.

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
