# Peer review of "Oncobiomics: Leveraging Microbiome Translational Research in Immuno-Oncology for Clinical-Practice Changes"

_biomolecules, 2025, doi:10.3390/biom15040504_

Round 1
Reviewer 1 Report
Comments and Suggestions for Authors
General comments: This authoritative review of translational studies in immuno-oncology (I-O) is focused on specific interactions of microbiota on anti-cancer immune response and response to immunotherapy. A wide range of factors and potential biomarkers are described and examined in the context of emerging pre- clinical and clinical data indicating that antibiotics negatively affect I-O therapies, that microbes can improve efficacy of treatment and can enhance immune response against some cancers. The review provides a comprehensive summary and analysis of current information and new developments and makes a valuable contribution to the field. Vigorous editing such paragraphing and subtitles is needed to improve the transitions from clinical to preclinical, whole body to tissues, etc. and would strengthen this deeply scholarly and timely review.
Specific Comments and questions:
1. Introduction (Line 26…): The regulatory impact of tumor microenvironment (TME), as described in the first paragraph, is not clearly related to the microbiome or linked to the foundational concepts of Oncobiomics, as subsequently presented. Additional information could be added or the paragraph could be transferred to the section on Impact of Microbes on Immunosurveillance. Lines 35- 75 are more than sufficient as the Introduction.
2. The review is too tightly written and demands intense study; subjects are abruptly changed, conclusions are merged with data presentation, and weighed down with references. For example, Line 156-161, ‘Although sequencing methods’ should start a new paragraph.
3. The authors should consider using subheadings (cancer-associated microbial neoantigens, intratumoral microbiota, taxonomic biomarkers). Lines 169 and following: The studies of intratumoral microbiota are highly interesting and could be discussed under separate heading that includes TME.
4. Line 209 Development of diagnostic tools could be a main heading.
5. Line 222: Comments on toxicity and effects on metabolic pathways could be expanded and explained
6. The section on Comedications could be improved with tables for ATB and FMT to set the context and use of precise comments in the text. Lines 435: “Given the influence that β-adrenergic signaling-mediated stress has on microbial health in a cancer setting, the use of β-blockers has been proposed as a potential therapeutic strategy.” What effect? Which cancer?
7. Sections on Next-generation probiotics (NGP), Live biotherapeutic products (LBP) and Diet, prebiotics, and metabolites are overly discursive and need compression. Some material if compressed could be integrated into previous sections. Example: Line 582 and following: “Other preclinical evidence shows promising results of metabolic modulation in oncology”
Author Response
Stylistic, typographical, grammatical, and other minor revisions:
We have revised the main manuscript to correct grammatical errors and clarify points we believed required revision for clarity. We have also consolidated references and conformed them to the journal’s citation style. Furthermore, we have addressed the remarks addressed by the reviewers below.
Reviewer #1:
- Line 209 Development of diagnostic tools could be a main heading.
- Done
Major Revisions
Some comments by Reviewer #1 and Reviewer #3 required more extensive revisions to the manuscript. Below are our responses.
Reviewer #1:
- Introduction (Line 26…): The regulatory impact of tumor microenvironment (TME), as described in the first paragraph, is not clearly related to the microbiome or linked to the foundational concepts of Oncobiomics, as subsequently presented. Additional information could be added or the paragraph could be transferred to the section on Impact of Microbes on Immunosurveillance. Lines 35- 75 are more than sufficient as the Introduction.
- Thank you for your suggestion. We have moved this paragraph from the Introduction to the opening paragraph of the “Impact of Microbes on Immunosurveillance” section.
- The review is too tightly written and demands intense study; subjects are abruptly changed, conclusions are merged with data presentation, and weighed down with references. For example, Line 156-161, ‘Although sequencing methods’ should start a new paragraph.
- Per the reviewer’s suggestion, we have started a new paragraph at Line 156. Furthermore, we have revised the language in the (now) preceding paragraph to improve clarity and readability. Furthermore, the citations have been changed to the journal’s required citation format, which should additionally improve readability in heavily sourced sentences. We’ve also removed some citations that ultimately seemed unnecessary (e.g., web-link citations associated with NTC codes for clinical trials) and consolidated duplicates of the same citations.
- The authors should consider using subheadings (cancer-associated microbial neoantigens, intratumoral microbiota, taxonomic biomarkers). Lines 169 and following: The studies of intratumoral microbiota are highly interesting and could be discussed under separate heading that includes TME.
- We have combined the sections entitled “Impact of microbes on immunosurveillance” and “The clinical relavence of extraintestinal microbiota” into one section, which was then broken into several subheadings to highlight the key themes.
- Line 222: Comments on toxicity and effects on metabolic pathways could be expanded and explained
- We have expanded the disucssion about these effects as follows:
Additionally, gut microbiota composition has been correlated with toxicity to both CAR-T cell therapies [90] and ICI [91,92], specifically the incidence and severity of IrAEs. Moreover, dysbiosis has been characterized by metabolic disruptions, notably reduced production of SCFAs, elevated levels of inflammatory cytokines, and compromised gut barrier integrity.
- The section on Comedications could be improved with tables for ATB and FMT to set the context and use of precise comments in the text.
- Thank you for the suggestion. We have incorporated a graphical table into a new Figure 2 that should add clarity to this section.
- Lines 435: “Given the influence that β-adrenergic signaling-mediated stress has on microbial health in a cancer setting, the use of β-blockers has been proposed as a potential therapeutic strategy.” What effect? Which cancer?
- We have added text and clarified the paragraph to better explain this point.
- Sections on Next-generation probiotics (NGP), Live biotherapeutic products (LBP) and Diet, prebiotics, and metabolites are overly discursive and need compression. Some material if compressed could be integrated into previous sections. Example: Line 582 and following: “Other preclinical evidence shows promising results of metabolic modulation in oncology”
- We have moved the sentences about antibiotic effects on bile acid metabolism to a section that focuses on the discussion on MAdCAM-1. We’ve also focused and expanded the discussion about prebiotics at the end of the section.

Reviewer 2 Report
Comments and Suggestions for Authors
Comments to the authors:
The authors present a review-style manuscript addressing the impact of the microbiome on immuno-oncology, addressing points of interest such as the role of the microbiota through what is currently known as “ONCOMIOCS”, postulating that the possibility of manipulating the microbiota could reinforce anti-cancer responses. It is important to have diagnostic tests for intestinal dysbiosis to improve therapeutic strategies.
The authors address the relationship of microbes in the function of immunosurveillance and highlight the information of certain molecules of the immune response, including the intracellular adhesion molecule mucosal addressin, which participates in the recruitment of leukocytes to inflamed tissues such as mucosa and intestinal microbiota, discussing and concluding that efforts still need to be analyzed and developed to guide the assessment of the anti-tumor effects of microbiota.
In the current submission, the authors should improve the presentation of their manuscript to highlight the importance of their revision, which allows me to comment on the following points:
- It is important to correct your table 1 that refers to the clinical guidelines for fetal microbiota transplantation in oncoimmunology of solid tumors. It is not aligned. The words, for example, route of administration, are short. Surely, when uploading your manuscript, the paragraphs were moved. It is important that the authors review them. Also, the indications of the meanings of the abbreviations in lines # 483 to # 496
- In the acknowledgements section line # 630 to # 654 I suggest correcting the paragraph spacing
- Correct bibliographic references to the journal format, lines #655 to 1214
Author Response
Stylistic, typographical, grammatical, and other minor revisions:
We have revised the main manuscript to correct grammatical errors and clarify points we believed required revision for clarity. We have also consolidated references and conformed them to the journal’s citation style. Furthermore, we have addressed the remarks addressed by the reviewers below.
Reviewer #2:
- It is important to correct your table 1 that refers to the clinical guidelines for fetal microbiota transplantation in oncoimmunology of solid tumors. It is not aligned. The words, for example, route of administration, are short. Surely, when uploading your manuscript, the paragraphs were moved. It is important that the authors review them. Also, the indications of the meanings of the abbreviations in lines # 483 to # 496
- Done
- In the acknowledgements section line # 630 to # 654 I suggest correcting the paragraph spacing
- Done
- Correct bibliographic references to the journal format, lines #655 to 1214
- Done
Reviewer 3 Report
Comments and Suggestions for Authors
The review article deals with microbiota dysfunction as a potential factor of carcinogenesis and cancer progression in terms of systemic antitumor immune responses and ICI efficiency. In general, these events are still poorly studied, the findings are controversial and heterogenic. Therefore, collaborative networks would be useful in order to identify microbiota-related biomarkers and develop novel microbiota-based strategies, reduce cancer risks and improve clinical outcomes in patients. Different approaches with pre-, probiotics and diet components are discussed. The authors refer a number of useful articles on this topic. Being written in attractive manner, this comprehensive review contains original ideas, thus being of interest to broad readership in the field.
Remarks:
Line 39: instead of ‘realization’, one could use, e.g., ‘concept’ or ‘hypothesis’
Line 80: Throughout the text, a harmful role of Enterocloster spp is discussed, as proven in several studies. Meanwhile, a negative prognostic role of Enterobacterales is also well known, especially for septic conditions in immunocompromised patients. One should, at least, mention this family as an example of post-ATB bacterial overgrowth associated with life-threatening problems.
Line 117-119: ‘gut microbiome may enhance immune surveillance by increasing …. cross-reactivity between microbial peptides and tumor-associated antigens…’. - One could suggest here some potential adverse effects, i.e., triggering autoimmune processes due to molecular mimicry’ E.g., such adverse effects are observed upon clinical use of ICI in cancer …
Line 163: ‘scientific question’ – ‘research tasks’ is more preferable here
Line 178: ‘mouse models’ – the term ‘murine models’ is more appropriate
Line 228: ‘We and others’ may be, for example, replaced by ‘we, together with our colleagues’…
Lines 235-237: The MGS (metagenomic sequencing) showed that signature A was ‘associated with various diseases’, whereas signature B was ‘positively associated with disease’: please specify which diseases are meant (in brief), or skip this passage.
Line 267: Does ‘Abnormal A. muciniphila abundances’ mean ‘lower relative contents’?
Line 275: ‘health individuals‘ should be ‘healthy…’
Line 325: The section about host-derived fecal miRNAs could be started from new line.
Line 367: ‘Immune responses … deserves’ replace by ‘….. deserve’
Line 468: Indeed, the good FMT responders ‘shift towards their donor’s gut microbial composition’ – it may also interpreted as a ‘shift towards normal microbiota ranges’, since the routine 16S-NGS mostly assesses biodiversity at the level of microbial genera and species.
These remarks are mostly, stylistic, to make the text more clear and understandable.
Comments on the Quality of English Language
Minor copy editing is required
Author Response
Stylistic, typographical, grammatical, and other minor revisions:
We have revised the main manuscript to correct grammatical errors and clarify points we believed required revision for clarity. We have also consolidated references and conformed them to the journal’s citation style. Furthermore, we have addressed the remarks addressed by the reviewers below.
Reviewer #3:
- Line 39: instead of ‘realization’, one could use, e.g., ‘concept’ or ‘hypothesis’
- We have used the word ‘concept’ rather than ‘realization’.
- Line 178: ‘mouse models’ – the term ‘murine models’ is more appropriate
- All instances of ‘mouse models’ has been revised to ‘murine models’.
- Line 228: ‘We and others’ may be, for example, replaced by ‘we, together with our colleagues’…
- Done
- Line 275: ‘health individuals‘ should be ‘healthy…’
- Done
- Line 325: The section about host-derived fecal miRNAs could be started from new line.
- Done
- Line 367: ‘Immune responses … deserves’ replace by ‘….. deserve’
- Done
Major Revisions
Some comments by Reviewer #1 and Reviewer #3 required more extensive revisions to the manuscript. Below are our responses.
Reviewer #3
- Line 117-119: ‘gut microbiome may enhance immune surveillance by increasing …. cross-reactivity between microbial peptides and tumor-associated antigens…’. - One could suggest here some potential adverse effects, i.e., triggering autoimmune processes due to molecular mimicry’ g., such adverse effects are observed upon clinical use of ICI in cancer …
- To highlight this potential for IrAEs, we have revised the text as follows:
Collectively, these findings also support the argument that a highly diverse gut microbiome may enhance immune surveillance by increasing the potential for cross-reactivity between microbial peptides and tumor-associated antigens, including neoantigens. However, cross-reactivity between gut microbial antigens and tumor neoantigens suggests a similar potential between gut microbes and autoantigens [39]. Such interactions could increase the incidence of IrAEs following ICI therapy and require further study.
- Lines 235-237: The MGS (metagenomic sequencing) showed that signature A was ‘associated with various diseases’, whereas signature B was ‘positively associated with disease’: please specify which diseases are meant (in brief), or skip this passage.
- Thank you for your comment. As suggested, we have revised the text to now explicitly (and briefly) list the specific diseases associated with microbiome signatures.
- Line 267: Does ‘Abnormal A. muciniphila abundances’mean ‘lower relative contents’?
- The reviewer refers to the description of muciniphila abundance discussed in the subsection 'Fecal markers.' By 'abnormal,' we specifically mean either the absence of A. muciniphila or its excessively high relative abundance (greater than 4.799%). Both extremes correlate with poorer clinical outcomes, immunosuppressive gut microbiota, and increased host exfoliate transcriptome. We have clarified this point explicitly in the revised manuscript to specify that 'abnormal' refers precisely to values outside the defined healthy range—either too low or excessively high.
Rebuttals
While we thank the reviewers for their suggestion, and we incorporated most of them, there were some points that we did not agree with. Please find our responses below.
Reviewer #3:
- Line 80: Throughout the text, a harmful role of Enterocloster spp is discussed, as proven in several studies. Meanwhile, a negative prognostic role of Enterobacteralesis also well known, especially for septic conditions in immunocompromised patients. One should, at least, mention this family as an example of post-ATB bacterial overgrowth associated with life-threatening problems.
- The text being referred to is:
ATB leads to reduced microbial (both fungal and bacterial) diversity and a compositional shift dominated by harmful Enterocloster spp. and Hungatella hathewayi [15,26,27], which negatively impact host metabolism and immunity [26,27]. For instance, our laboratory has demonstrated that the “blooming” of Enterocloster spp. following ATB cessation disrupts the expression of ileal mucosal addressin cell adhesion molecule-1 (MAdCAM-1) through perturbation of bile acid metabolism [26,27]. The downregulation of MAdCAM-1 on the luminal side of high-endothelial venules in the gut mucosa consequently allows the migration of (normally) gut-resident immunosuppressive a4b7+ regulatory Th17 (Tr17) cells towards extraintestinal tumors, consequently promoting an immunosuppressive TME and leading to cancer progression and treatment resistance [26].
We thank the reviewer for highlighting the established negative prognostic impact of post-antibiotic overgrowth of Enterobacterales in immunocompromised patients. While it is true that blooms of Enterobacterales are linked to life-threatening sepsis, our text focuses specifically on the mechanisms by which Enterocloster spp. disrupt bile acid metabolism and downregulate MAdCAM-1, leading to the migration of immunosuppressive Tr17 cells into extraintestinal tumors. Although discussing Enterobacterales as an additional example of post-antibiotic dysbiosis could provide broader context, it does not support the message we are trying to communicate.
- Line 163: ‘scientific question’ – ‘research tasks’ is more preferable here
- We appreciate the reviewer's suggestion to replace 'scientific question' with 'research tasks' in line 163. However, we maintain that 'scientific question' better captures the intended meaning within this context. Specifically, our point emphasizes that the selection of sequencing methods should be guided primarily by the underlying scientific inquiry, rather than merely operational tasks. We have added language in the passage to enhance this point.
- Line 468: Indeed, the good FMT responders ‘shift towards their donor’s gut microbial composition’ – it may also interpreted as a ‘shift towards normal microbiota ranges’, since the routine 16S-NGS mostly assesses biodiversity at the level of microbial genera and species.
- The reviewer is specifically referring to the passage:
FMT trials highlight the clinical relevance of the ecological taxonomic distance between the fecal microbiota of FMT donors and recipients, in addition to the engraftment of relevant microbial species (Baruch et al., 2021; Davar et al., 2021; Routy et al., 2023). Patients who benefit from FMT experience a gut microbiome shift toward their donor’s gut microbial composition (Baruch et al., 2021; Davar et al., 2021; Routy et al., 2023).
We thank the reviewer for their suggestion but maintain that our wording in this passage is accurate. First, the cited studies explicitly demonstrate that patients who respond favorably to FMT exhibit a shift in their gut microbiota composition toward that of their donors. Second, the term "normal microbiota ranges" is currently ambiguous, as there is no consensus for what constitutes a "normal" microbiome composition. This in itself is a major question in the field that requires further research.

Round 2
Reviewer 1 Report
Comments and Suggestions for Authors
This outstanding and authoritative review provides new guidance for a critical field in cancer research. The authors explore specific relationships and interactions that clarify how microbiota- based strategies could be examined and used to assess and improve outcomes for cancer management and treatment.